# Assessing Heat Resistance and Selecting Heat-Resistant Individuals of Largemouth Bass (*Micropterus salmoides*) with Tiered Thermal Exposure

**DOI:** 10.3390/ani15020128

**Published:** 2025-01-08

**Authors:** Haijie Chen, Hui Qiao, Zhicheng Xv, Guili Song, Shuning Liu, Cheng Luo, Yong Long, Shimei Lin

**Affiliations:** 1College of Fisheries, Southwest University, Chongqing 400715, China; haijie0602@163.com (H.C.); xuzhicheng3010@163.com (Z.X.); 2Key Laboratory of Breeding Biotechnology and Sustainable Aquaculture, Institute of Hydrobiology, Chinese Academy of Sciences, Wuhan 430072, China; qiaohui1@zjou.edu.cn (H.Q.); guilisong@ihb.ac.cn (G.S.); liusn29@163.com (S.L.); 3National Engineering Research Center for Marine Aquaculture, Zhejiang Ocean University, Zhoushan 316022, China; 4Xiaogan Academy of Agricultural Sciences, Xiaogan 432100, China; 15072571717@163.com

**Keywords:** heat resistance, largemouth bass, tiered heat exposure, critical thermal maximum, lethal cumulative temperature

## Abstract

Largemouth bass (LMB) is economically important but highly sensitive to heat stress. Breeding heat-resistant LMB is of big significance in the face of global warming. However, there still lacks an applicable method to efficiently assess the heat resistance of LMB individuals. In this study, we developed a tiered exposure method where the temperature was increased step-wise and the heat resistance of fish was quantified as the lethal cumulative temperature (LCT). Largemouth bass juveniles could be classified as sensitive or resistant to heat stress based on the LCT measurements. Higher degrees of tissue damage and cell apoptosis were found in the livers of the heat-sensitive individuals. Differential expressions of genes involved in the endoplasmic reticulum stress response and apoptosis were also detected in the livers of sensitive and resistant fish. Additionally, LMB juveniles were found to be more resistant than adults and extremely heat-resistant individuals were successfully selected using the tiered heat exposure method. Our method and data are valuable for understanding the thermal biology of LMB and breeding heat-resistant LMB varieties.

## 1. Introduction

With the continuous growth of the global population, the demand for aquatic products as an important food source of high-quality protein is expected to keep increasing [1]. In the past few decades, aquaculture has become one of the fastest-growing food industries in the world. It is expected that by 2030, aquaculture will contribute 53% of fishery products [2]. However, the stability and sustainability of aquaculture development are currently facing challenges due to global warming [3]. Over the past thirty years, human activities have led to an great increase in atmospheric greenhouse gas emissions, causing the global average temperature to rise by 0.2 °C every decade [4]. Furthermore, the global average temperature was projected to continue rising in the coming centuries [5]. The rise of water temperature due to global warming poses a significant threat to the health of aquatic animals and the sustainable development of aquaculture [6].

Temperature is the master factor that determines all life activities of fish, such as feeding [7], growth [8], reproduction [9] and metabolism [10]. Fish face significant challenges when exposed to water temperatures beyond their tolerance limits [11]. Largemouth bass (LMB, *Micropterus salmoides*) is widely cultured around the world due to its attractive traits, such as rapid growth, adaptability to handling, being free of intermuscular bones and fillet taste [12]. The aquaculture of LMB grew very fast, especially in China. According to the *China Fisheries Statistical Yearbook*, the farmed output of LMB reached 0.88 million tons in 2023. However, the aquaculture of LMB has been badly impacted by the heatwaves in the main cultivation regions, such as China, in recent years. In addition to a decrease in growth, extended exposure of LMB to heat stress results in tissue damage, functional impairments of the organs and susceptibility to pathogen infection [13,14,15,16].

Heat stress has been reported to induce endoplasmic reticulum stress and promote apoptosis in the liver of LMB via the IRE1α/TRAF2/ASK1/JNK pathway [14,17]. The degree of heat-induced cell apoptosis in fish tissues was proportional to the severity of the stress, and a TUNEL assay was extensively applied to investigate the heat-induced cell apoptosis in fish [16,18]. Comparative transcriptomic analysis of the liver of the LMB revealed that the ECM-receptor interaction pathway was one of the most dramatically changed pathways affected by heat exposure [19]. Gene alternative splicing events elicited by heat exposure were identified from the gills of LMB, where the genes that underwent heat-induced differential splicing were associated with immune-related pathways, such as necroptosis, apoptosis and C-type lectin receptor signaling [20]. Heat stress was reported to induce enteritis in fish, and the effects of acute heat stress on the long noncoding RNAs (lncRNAs) associated with intestinal immunity of LMB were characterized [21]. Acute heat exposure triggered liver transcriptome and intestine microbial remodeling in LMB, and the microbial biomarkers/genes that monitor the health status of fish were identified [22]. Although much is known about the effects of heat stress on the physiology and biological processes of LMB, the molecular mechanisms underlying the heat sensitivity of LMB remain unknown.

Identifying the master genes that determine the heat resistance of LMB and elucidating the underlying molecular mechanisms are critical for breeding heat-resistant LMB strains. Furthermore, developing feed additives aimed at enhancing heat resistance is also a good strategy for the sustainable development of aquaculture. All of these research fields require the efficient, accurate and high-throughput assessment of the heat resistance of individuals. The critical thermal maximum (CTmax) [23] and static heat exposure [24] are the two commonly used methods for evaluating the thermal resistance of fish [25]. When the water temperature is linearly increased, the temperature at which an individual loses the ability to maintain equilibrium is regarded as its CTmax [26]. The CTmax method has been widely used to evaluate the thermal resistance of aquatic animals [27] because it is fast, easy to operate and does not rely on observing the mortality of the experimental animals [28]. For the static heat exposure method, the thermal resistance of animals is indicated by the survival time under the lethal temperature [29,30].

In this study, the heat resistance of juvenile LMB was initially assessed using two commonly used methods. The results revealed limitations in both methods, as they were unable to effectively and accurately differentiate the heat resistance of individual fish. To address this, we developed a tiered exposure method that combined the two classical approaches. This method involved increasing the temperature in a step-wise manner and quantifying the heat resistance of the fish as the lethal cumulative temperature (°C·h). By employing this method, we were able to clearly distinguish between heat-sensitive and heat-resistant individuals within the LMB population. Furthermore, the sensitive individuals exhibited more severe tissue damage and cell apoptosis in the liver compared with the resistant individuals. The tiered exposure method was also utilized to compare the heat resistance of LMB juveniles and adults, as well as to identify extremely heat-resistant LMB individuals. Our findings shed new light on the thermal biology of fish and offer potential applications in breeding heat-resistant LMB strains.

## 2. Materials and Methods

### 2.1. Fish Maintenance

Except where otherwise stated, the experimental LMB used in this study were obtained from fish seed farms and acclimated for 2 to 4 weeks in the laboratory before treatment. The fish were maintained in round aquaria (diameter 1 m) supplied with recycled water. The water parameters, including dissolved oxygen > 5 mg/L, ammonia nitrogen < 0.1 mg/L and nitrite nitrogen < 0.05 mg/L, were maintained during fish cultivation. The fish were fed with a commercial extruded feed obtained from Fujian Tianma Science and Technology Group Co., Ltd. (Fuzhou, China), twice daily (9:00 a.m. and 5:00 p.m.) to satiation. The nominal feed composition was crude protein 53%, crude lipid 5%, crude fiber 3% and crude ash 15%. The source, development stage, body size and rearing temperatures of different batches of experimental fish used in this study are shown in Table 1.

### 2.2. Determination of Critical Thermal Maximum

The critical thermal maximum (CTmax) was measured for two batches of LMB juveniles (Table 1, CTmax). After being fasted for 24 h, the fish to be tested were placed into the constant temperature water tank (filled with 100 L water) and maintained at 28 °C for 24 h, as previously reported [31]. Nine to eleven fish were analyzed each time. The fish were transferred into the chamber of a PC200 A40 ARCTIC Refrigerated Circulator (Thermo Scientific, Waltham, MA, USA) filled with 12 L water and acclimated at 28 °C for 1 h. After this, the water temperature was increased from 28 °C with a constant rate of 0.3 °C/min [32]. The temperature at which an individual demonstrated a loss of equilibrium (LOE) for more than 30 sec was regarded as its CTmax. The fish that reached CTmax were taken out and the body weight and standard length were measured. The condition factor (K) of individuals was calculated using the formula K = 100 × W/L^3^, where W and L represent the body weight (g) and standard length (cm), respectively.

### 2.3. Static Heat Exposure

Four batches of LMB juveniles were subjected to static heat exposure (Table 1, static exposure). In the static exposure experiments, the fish were directly transferred from the cultivation aquaria to the exposure tanks with water heated to the desired temperature [16,22]. The exposure tanks were filled with 600 L water. Submerged heaters (A/C 220V) obtained from Yameiguangyi (Qingdao, China) were used for the water temperature control. Each heater was connected to a thermostat (811-H, Yameiguangyi, Qingdao, China) that could shut off the heater when the desired temperature was reached. The fish were not fed during the static heat exposure. The water was continuously aerated and about 1/10 of the water (preheated to the desired temperature) was changed every day to prevent the deterioration of the water quality. The fish were checked every 6 h and the death of fish was judged by a loss of response to touch and the body being stiff. The dead fish were removed, the survival times were recorded, and the body weights and standard lengths were measured.

### 2.4. Tiered Heat Exposure and Calculation of Lethal Cumulative Temperature (LCT)

The facility for the tiered heat exposure is illustrated in Appendix A. It consisted of an HXL-M030 thermostat (cooling or heating volume: 2000–3200 L, temperature controlling range: 2–50 °C) produced by Guangdong Haojia Refrigeration Equipment Co., Ltd. (Guangzhou, China), a filtering tank, a clean water tank, four exposure aquaria (diameter 1 m, filled with 600 L water), two pumps, connecting pipes and valves. The temperature of the water in the exposure aquaria was measured and recorded using a temperature recorder produced by Beijing Saiouhuachuang Technology Co., Ltd. (Beijing, China).

LMB juveniles obtained from 3 different farms (Table 1, tiered exposure, Experiment#1) were used to test the tiered exposure method. The fish were fasted for 24 h before being transferred into the exposure aquaria (preheated to 28 °C). After being acclimated to 28 °C for 24 h, the temperature was increased 2 °C per step from the rearing temperature (28 °C) to the non-lethal temperature (34 °C), followed by 0.5 °C per step above 34 °C. Each step lasted one day. The temperature setting was changed at 9:00 a.m. every day. Other procedures, such as the fish inspection and death judgement, were the same as for the static heat exposure.

The data exported from the temperature recorder were analyzed to evaluate the accuracy of the temperature control. During the experiment, the desired temperatures were reached in 120 min after changing the settings of the thermostat from different starting temperatures, despite the temperature increment being 2 °C (Appendix A) or 0.5 °C (Appendix A). The temperature was increased from 28 to 39 °C in a step-wise manner during the experiment (Appendix A). The standard deviation of the measured temperature ranged from 0.03 to 0.23 °C, the difference between the maximum and minimum temperatures in each day ranged from 0.2 to 0.4 °C, and the deviation between the desired temperature and the mean measured temperature ranged from −0.19 to 0.10 °C (Appendix A). These data indicate that the experimental facilities used in this study could stably control the water temperature.

The lethal cumulative temperature (LCT, °C·h) of each individual was calculated with the following formula:$$LCT=\sum_{i=1}^{\left(n-1\right)}\left(Ti-T0\right)\times 24+(Tn-T0)\times m$$

T0: the reference temperature (℃);

Ti: temperature on the i-th day;

Tn: the death temperature (℃);

n: the survival time (days);

m: the survival time at the death temperature (hours).

Since the experimental fish did not die when the water temperature was below 34 °C, we used 34 °C as the reference temperature (T0). Because the fish were checked every 6 h, we divided each day into four 6 h periods. For example, if a fish died at 12 h after the water temperature reached 35 ° C, its LCT was calculated as (34.5 − 34) × 6 × 4 °C·h + (35 − 34) × 6 × 2 °C·h = 24 °C·h. A table for the death temperature, survival time and corresponding LCT was compiled to facilitate the LCT calculation (Appendix A). The LCT values can be obtained by searching the table according to the death temperature and survival time. The table can be adjusted if the parameters, such as reference temperature (T0), temperature increment and observation frequencies, are changed.

### 2.5. Identification and Characterization of Heat-Sensitive LMB Individuals

Tiered heat exposure was repeated as described above using another batch of LMB juveniles (Table 1, tiered exposure, Experiment#2) to identify the sensitive and resistant individuals. When the water temperature reached 35.5 °C and was maintained for 24 h, the sensitive and resistant fish were easy to distinguish. The sensitive fish floated up to the water surface, had an increased breathing frequency and had a decreased response to netting stress, while the resistant ones were free of these behavioral abnormalities. Two sensitive and resistant individuals were collected from each tank. The controls were sampled before the heat exposure. In total, 6 individuals were sampled for each experimental group. The fish were euthanized by immersion in 250 mg/L MS-222 (St. Louis, MO, USA) before dissection. The brain and liver tissues were individually collected and subjected to histology and gene expression assays.

#### 2.5.1. Histological Analysis and TUNEL Staining

Histological analysis and TUNEL (terminal deoxynucleotidyl transferase dUTP nick end labeling) staining were performed, as previously described [16]. Briefly, the brain (mesencephalon) and liver tissues were fixed in 4% paraformaldehyde, dehydrated with alcohol gradients, embedded in paraffin and sliced to 4 µm, and the sections were stained with H&E (hematoxylin and eosin). The stained sections were scanned using an Aperio VERSA Brightfield, Fluorescence and FISH Digital Pathology Scanner from Leica (Wetzlar, Germany). Three consecutive sections were analyzed for each sample. The One-Step TUNEL Apoptosis Assay Kit from Beyotime (Shanghai, China) was used to detect apoptotic cells in the tissue sections (consecutive sections of the same samples used for histological analysis), following the manufacturer’s instructions. Images for the sections were taken using an SP8 fluorescent confocal microscope from Leica (Wetzlar, Germany). The photos were analyzed using ImageJ2 [33] to count the nuclei and apoptotic cells. The severity of cell apoptosis was expressed as a ratio of the apoptotic cells to the total number of nuclei.

#### 2.5.2. Total RNA Isolation and Quantitative PCR

The whole-brain and liver samples were subjected to total RNA extraction using the TRIzol reagent from Thermo Fisher Scientific (Waltham, MA, USA) according to the manufacturer’s instructions. The integrity of the RNA samples was assessed by agarose gel electrophoresis. The RNA concentration was measured by using a Q5000 UV–Vis spectrophotometer (Quawell, San Jose, CA, USA). The EasyScript One-step gDNA Removal and cDNA Synthesis SuperMix from TransGen (Beijing, China) was used for the cDNA synthesis. Real-time quantitative PCR (qPCR) assays were performed using a CFX Duet Real-Time PCR System from Bio-Rad (Hercules, CA, USA). The reagents, program and method were the same as the previous study [16]. The sequence, amplicon size and efficiency of the qPCR primers are listed in Appendix A. The geometric average of the expressions of *actb2* and *eef1g* was used as the normalization factor for qPCR data analyses.

### 2.6. Application of the Tiered Exposure Method

#### 2.6.1. Comparing Heat Resistance of LMB Juveniles and Adults

To compare the heat resistance of LMB adults and juveniles, winter-aged adults of LMB were obtained from two fish farms in Wuhan, China. The adult fish were caught from the ponds with a cast net, transferred into the laboratory aquaria and allowed to acclimate to the experimental condition for 2 weeks. The juveniles were maintained under the same condition. The number, body size and rearing temperatures of the adult and juvenile LMB are displayed in Table 1, tiered exposure, Experiment#3. The fish were subjected to tiered heat exposure, as described above.

#### 2.6.2. Selecting Heat-Resistant Individuals from the LMB Population

To select the heat-resistant individuals of LMB, the fish introduced in Table 1, tiered exposure, Experiment#4 were randomly assigned to the 4 tanks of the heat exposure system. Tiered heat exposure was conducted and the LCT of the dead fish was calculated as described above. Heating was stopped when the water temperature reached 38 °C and maintained for 6 h. At the end of the treatment, the survivors were counted and allowed to recover in the exposure tank. The water cooled naturally to room temperature. The fish that died during the recovery phase were taken out and subjected to body measurements. Finally, the successfully recovered fish were transferred into the rearing aquarium and subjected to body measurements after being anesthetized with 80 mg/L MS-222.

### 2.7. Statistics

Statistical analyses were performed using GraphPad Prism (v9.3.1). Simple linear regression was performed to analyze whether there was a significant correlation between the CTmax and body weight or condition factor and between the LCT and body weight of the experimental fish. The difference in gene expression between the experimental groups was analyzed by an independent samples *t*-test. The differences in death temperature and LCT between different experimental groups were analyzed by one-way ANOVA followed by Tukey’s multiple comparisons test. A *p*-value less than or equal to 0.05 served as the threshold for statistical significance.

## 3. Results

### 3.1. CTmax of Largemouth Bass Juveniles

The CTmax was measured for two batches of LMB juveniles with different average body weights (Table 1). The first batch (19 fish) had a CTmax that ranged from 39.1 to 39.9 °C (Appendix A). The second batch (31 fish) had quite similar values that ranged from 39.0 to 40.0 °C (Figure 1A). More than half of the CTmax values for the second batch of fish were concentrated between 39.4 and 39.8 °C (Figure 1A). For the experimental population that had a body weight between 4.2 and 16.6 g, no significant correlation was identified between the CTmax and body weight (*p* = 0.319, Figure 1B). Furthermore, the condition factor of the fish also had no significant correlation with the CTmax (*p* = 0.666, Figure 1C). In summary, the CTmaxs of different fish were quite similar and could not clearly differentiate the heat resistance of the individuals in the same population.

### 3.2. Survival Times of Largemouth Bass upon Static Heat Exposure

In total, four static exposure experiments were conducted to assess the heat resistance of the LMB juveniles. In the first experiment, the fish that were cultured between 23 and 25 °C and had an average body weight of 17.58 ± 5.00 g were immediately transferred from the rearing temperature to 28, 30, 32, 34 or 36 °C. The results indicate that all the fish exposed to 36 °C died in 2 d, while no mortality was found for the other experimental groups (Figure 2A). In the second experiment, the fish were reared between 24–26 °C before heat exposure and had an average body weight of 100.62 ± 23.31 g. The fish were exposed to the same temperature gradients as in the first experiment and only minimal mortality (15%) was found for the highest-temperature group (36 °C) in 14 d (Figure 2B). In the third experiment, the fish that were cultured under 24–27 °C and weighed 96.01 ± 17.53 g were immediately exposed to 36.5, 37 or 38 °C. All the fish exposed to 37 or 38 °C died in 6 h, the first inspection time point after the beginning of exposure; all the fish exposed to 36.5 °C died in 18 h and 80% of the mortality occurred between 12 and 18 h after the beginning of the exposure (Figure 2C). In the fourth experiment, the fish that weighed 45.12 ± 10.47 g and were reared under 26–28 °C were exposed to 36, 36.5 or 37 °C. The results indicate that all the fish exposed to 37 °C died in 12 h, all the fish exposed to 36.5 °C died in 48 h and all those exposed to 36 °C died in 96 h (Figure 2D).

Together, the survival times of the fish in the static exposure experiments were intimately dependent on the rearing and treatment temperatures. The difference between the survival and death temperatures was very small. A minimal increase in the exposure temperature (0.5 °C) could lead to mass mortality in a short period of time. Therefore, it is hard to predetermine an applicable exposure temperature to differentiate the heat resistance of the individuals in a population.

### 3.3. Lethal Cumulative Temperature of Largemouth Bass upon Tiered Heat Exposure

The LMB juveniles obtained from three different breeding farms were subjected to tiered heat exposure. The death temperatures of the fish upon exposure to the continuously intensified heat stress ranged from 35.5 to 39 °C (Figure 3A). The LCT values of the fish were from 39 to 594 °C·h (Figure 3B). While multiple individuals died at the same temperature (Figure 3A), the LCTs were more scattered (Figure 3B). Interestingly, a bimodal distribution pattern was identified for both the death temperature and LCT, where a markedly lower data density was identified around 37 °C and 200 °C·h for the death temperature and LCT (indicated by the red lines in Figure 3A,B). These data indicate that the experimental fish could be classified into heat-sensitive and -resistant subgroups. Again, no significant correlation was identified between the body weight and LCT of the fish (*p* = 0.283, Figure 3C).

### 3.4. Heat-Sensitive Fish Demonstrated Tissue Damage and Apoptosis in the Liver

Sensitive and resistant individuals were identified from a new batch of LMB juveniles subjected to tiered heat exposure. Consistent with the behavioral observations, the H&E-stained liver sections of the heat-sensitive fish demonstrated a large numbers of vacuoles, indicating extensive tissue damage (Figure 4A). The liver sections of the heat-resistant fish had no obvious vacuoles but demonstrated deeper hematoxylin staining than those of the untreated controls (Figure 4A), suggesting catabolism of the nutrients deposited in the hepatocytes. However, no obvious changes were found for the brain (mesencephalon) sections of both the sensitive and resistant fish when compared with those of the controls (Figure 4A), indicating that the brain was less affected by the heat stress. Accordingly, while only minimal apoptotic cells could be seen in the mesencephalon sections, large numbers of apoptotic cells (Figure 4B) and a significantly higher ratio of apoptotic cells were found for the liver sections of the heat-sensitive fish (*p* < 0.05, Appendix A).

### 3.5. Expressions of the ER Stress-Response- and Apoptosis-Associated Genes

Except for *bax*, all the others were up-regulated in the brains of both the sensitive and resistant fish, but no significant difference was identified between the fish with different heat resistances (Figure 5). Only the *ire1* and *atf4* genes demonstrated consistent up-regulation in the brain and liver samples. The expressions of *perk*, *bcl2* and *casp3* were up-regulated in the brain but down-regulated in the liver upon heat exposure (Figure 5). Interestingly, the heat-sensitive fish had significantly lower expressions of *ddit3*/*chop* (*p* < 0.05) and *casp3* (*p* < 0.01) and higher expressions of *bax* (*p* < 0.01) in the liver (Figure 5). These genes can serve as molecular markers for the degree of heat-induced tissue damage and heat resistance of LMB.

### 3.6. Comparing Heat Resistance of Adult and Juvenile Largemouth Bass

The winter-aged adults of two different sources and the juveniles raised in the laboratory aquaria were subjected to tiered heat exposure. Compared with the two groups of adults, the juveniles had a significantly (*p* < 0.05) higher death temperature (Figure 6A). The juveniles also had a significantly (*p* < 0.05) higher LCT than the adults (Figure 6B). Although adult_S#2 had a higher average death temperature and LCT than adult_S#1, the differences were not significant (Figure 6A,B). Again, the LCT values were more scattered than the death temperatures. Multiple individuals that died at the same temperature had different LCTs (Figure 6A,B). Furthermore, a significant negative correlation was identified between the LCT and body weight (Figure 6C). Together, these data indicate that the juveniles of LMB had significantly stronger heat resistance than the adults.

### 3.7. Selection of Heat-Resistant Largemouth Bass from the Population

The fish subjected to selection could be classified as heat sensitive and resistant as well based on the LCT values (Figure 7A). Except for the dead ones, a total of 10 fish survived at the end of the heat exposure. Although no significant correlation was identified between the LCT and body weight of the dead fish (Figure 7B), the survivors had a significantly lower body weight than the dead ones (*p* < 0.05, Figure 7C). Finally, three fish resumed feeding and survived, and all the others died during recovery. The survival rate for the population subjected to heat selection was about 1% (3/263). Together, we established a procedure to select extremely heat-resistant individuals from the LMB population.

## 4. Discussion

Breeding heat-resistant fish strains and developing feed additives to enhance the heat resistance require accurate, efficient and high-throughput evaluation of the heat resistance. To assess the heat resistance of LMB, we measured the CTmax of the juveniles. Fish raised in the laboratory at temperatures of 26–28 °C had CTmax values that ranged from 39 to 40 °C, which was consistent with previous studies. Previous research reported that Florida LMB acclimated to 32 °C had an average CTmax of 39.2 °C with a temperature increase of 0.2 °C/min [34]. In another experiment, LMB juveniles acclimated to 30 °C had a CTmax around 38.5 °C [35]. Similar to previous studies, the standard deviation of CTmax for LMB was generally less than 0.5 °C [35], indicating a narrow range of values among individuals. However, the CTmax experiment, which requires a constant rate of water temperature increase, is limited to small-scale assessments and is not suitable for high-throughput evaluation. Despite its ease, rapidness and high repeatability, CTmax is more suitable for comparing the thermal resistance between species [28,36] but is not applicable for selecting heat-sensitive and heat-resistant individuals within LMB populations.

The static exposure method assesses the heat resistance of fish by measuring the survival time under a constant treatment temperature [29,30]. This method requires determining an optimal exposure temperature in advance. We conducted tests using various temperatures with multiple batches of LMB juveniles. The results indicate that finding a universally applicable temperature for fish raised under slightly different temperature ranges is challenging. Additionally, predicting the outcome of static exposure is difficult. In many cases, either minimal mortality occurred over a prolonged period or acute death happened quickly. This variability may have been due to the narrow range between survival and death during the acute exposure. Therefore, evaluating the heat resistance of individual LMB through static heat exposure is complex, as multiple exposure temperatures must be tested in advance to determine the optimal temperature for each batch of experimental fish.

To address the issues with CTmax and static exposure methods, we developed a tiered exposure method that combines dynamic and static approaches. In this method, the temperature is increased step by step with two different heating rates. A fast heating rate (2 °C per step) from the rearing temperature (28 °C) to a non-lethal high temperature (34 °C) reduces the heating time, while a slow heating rate (0.5 °C per step) above 34 °C accurately distinguishes the heat resistance levels of individuals. This method can be conducted in larger containers, as it does not require precise linear heating like CTmax. It also does not need a predetermined exposure temperature like static exposure. Therefore, the tiered exposure method is efficient, accurate and suitable for the high-throughput evaluation of fish heat resistance. The heat resistance of the fish was evaluated using the LCT measurement. This means that the higher the temperature that an individual can endure and the longer the survival time, the stronger its ability for heat resistance. The LCT measurement provides information on both the death temperature and survival time, allowing for differentiation between individuals that die at the same temperature based on the survival time. Additionally, the cumulative temperature was utilized to assess the impact of temperature fluctuations on the maturity of gonadal development [37], embryogenesis [38] and the growth of fish [39].

The LMB population could be classified into heat-sensitive and heat-resistant subgroups based on the LCT measurements. Consistently, histological and TUNEL assays revealed a higher degree of tissue damage and cell apoptosis in the liver of the sensitive fish. However, the brain of the sensitive fish was not affected by the heat stress. Furthermore, differential expressions of genes involved in the ER stress response and apoptosis were only found between the liver of the sensitive and resistant fish. Therefore, compared with the brain, the liver of the LMB was more susceptible to heat stress and played a more important role in determining the organism’s heat resistance. This is different from the effects of cold stress, as the Nile tilapia exposed to a lethal temperature demonstrated extensive damage in multiple tissues, including the brain and liver [31]. Heat-stress-induced apoptosis, oxidative damage and ER stress responses have been documented in the livers of multiple species [14,18,40]. Thermal-stress-induced hepatotoxicity was supposed to account for the death of *Labeo rohita* upon heat exposure [40].

Expressions of the ER stress-response- and apoptosis-associated genes were characterized to explore the roles of these pathways and genes in determining the heat susceptibility of LMB individuals. All the tested genes, except for *bax*, demonstrated a similar degree of up-regulation in the brain of both the sensitive and resistant fish. Since no obvious tissue damage and apoptosis signal were found in the brain sections of the heat-sensitive fish, the up-regulation of these genes may be related to the resilience of the brain tissue to heat stress. Considering the significantly lower expressions of the *chop* and *casp3* genes in the liver of the sensitive fish compared with those of the resistant ones, the expression levels of these genes may be positively related to the ability of cells to withstand the heat stress. In contrast, the up-regulation of the *bax* gene in the liver of the heat-sensitive fish suggests a negative correlation between its expression and heat resistance. The *ddit3*/*chop* gene encodes a basic leucine zipper (bZIP) transcription factor and plays critical roles in regulating cellular stress responses, glycolysis and autophagy [41,42,43]. Casp3 was shown to have a protective role in tissues of mice exposed to various chemical and environmental stresses by activating the antiapoptotic Akt kinase [44]. However, the roles of these genes in coping with heat stress remain to be further investigated.

Furthermore, there was no significant correlation between the LCT and body weight for the population with a relatively small variation in body size. These data suggest the presence of main-effect genes that determine the largest portion of variation in the LCT data. While the difference in LCT between individuals belonging to the same subgroup may be determined by small-effect genes or by the interactions between genetics and the environment. Changes in gene expression in the LMB tissues upon heat exposure were previously investigated and the heat-responsive genes were identified [15,16,19]. However, the functions of the heat-responsive genes in regulating the heat resistance of LMB remain to be further explored. In a recent genome-wide association study, 10 single nucleotide polymorphisms (SNPs) and 38 candidate genes were found to be significantly associated with robustness (assessed by the time needed to demonstrate an LOE (loss of equilibrium)) of LMB upon exposure to heat stress [45]. The tiered exposure method developed in this study can be used to test the roles of the candidate SNPs and genes in determining the heat resistance of LMB and other species.

Except for developing a method for assessing the heat resistance of fish, this study also revealed the higher heat sensitivity of LMB adults than juveniles. Therefore, when raising adult LMB, extra caution should be taken to avoid the adverse impacts of summer heatwaves. Finally, the successful selection of individuals with the strongest heat resistance from the population paves the way for breeding heat-resistant LMB varieties.

## 5. Conclusions

CTmax of the LMB juveniles raised at 26–28 °C was measured as 39–40 °C. Both the CTmax and static exposure methods failed to clearly separate the heat-sensitive and heat-resistant LMB individuals. The tiered exposure method addressed the issues of the CTmax and static exposure methods, whereby the temperature was increased in a step-wise manner, and the heat resistance of the fish was quantified with the lethal cumulative temperature (LCT). The LMB populations could be classified into heat-sensitive and heat-resistant subgroups based on the LCT values. The heat-sensitive individuals demonstrated severe tissue damage and cell apoptosis in the liver. Genes, including *ddit3*/*chop*, *casp3* and *bax*, were differentially expressed between the liver of the sensitive and resistant fish. The juveniles of LMB had a significantly higher LCT than the winter-aged adults. Furthermore, the most resistant individuals that could withstand extreme heat stress were successfully selected from the LMB population using the tiered exposure method.

## Figures and Tables

**Figure 1 animals-15-00128-f001:**
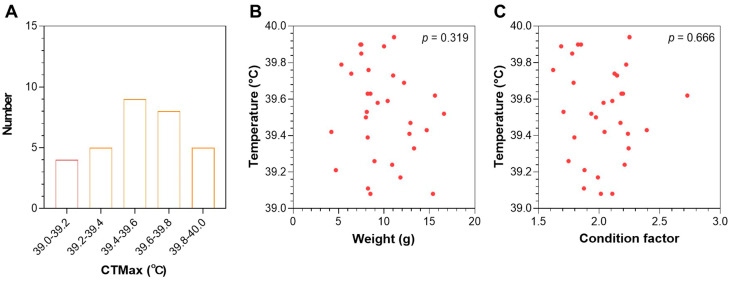
CTmax of the LMB juveniles. (**A**) Frequency distribution of CTmax (n = 31). (**B**) Correlation between CTmax and body weight. (**C**) Correlation between CTmax and condition factor.

**Figure 2 animals-15-00128-f002:**
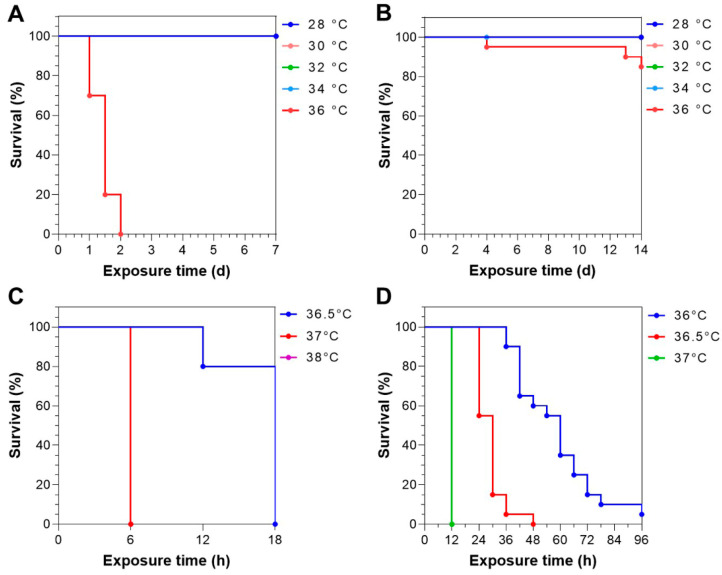
Survival curves of the LMB juveniles in the static heat exposure experiments. (**A**–**D**) Results of the first (**A**), second (**B**), third (**C**) and fourth (**D**) experiments. Fish number for each experimental group: (**A**) 10; (**B**) 20; (**C**) 10; (**D**) 20.

**Figure 3 animals-15-00128-f003:**
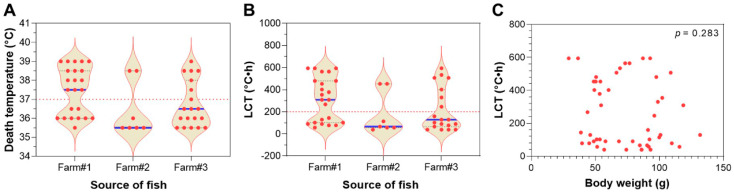
Death temperature and lethal cumulative temperature (LCT) of the LMB juveniles of different sources. (**A**) Death temperature of the fish (n = 23, 7 and 20). (**B**) LCT of the fish. The blue lines indicate medians. The red dashed lines indicate the bottleneck of the violin plots. (**C**) Correlation between the LCT and body weight.

**Figure 4 animals-15-00128-f004:**
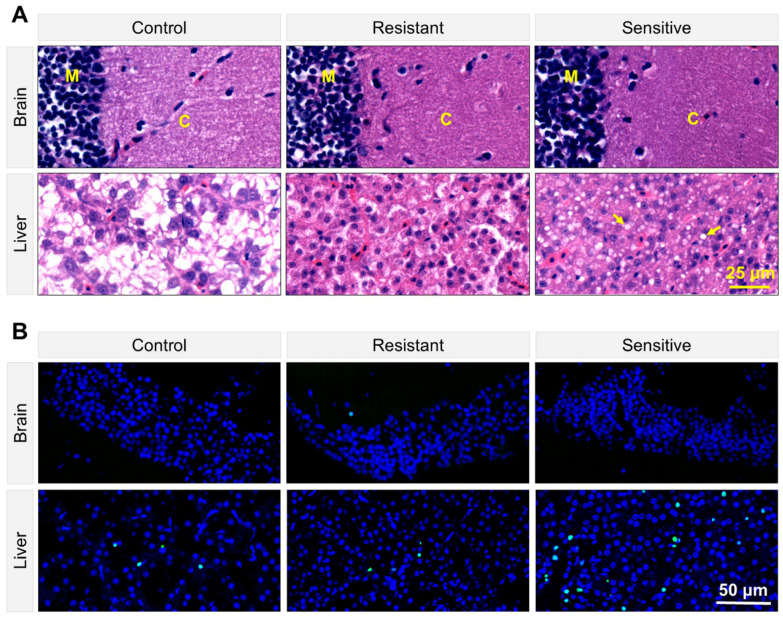
Photos of the H&E- and TUNEL-stained tissue sections. (**A**) Representative H&E-stained sections of the brain and liver tissues. Six sections of each experimental group were analyzed. C, cortex; M, medulla; arrow heads, vacuoles in the liver section. (**B**) TUNEL-stained sections of the brain and liver tissues. Three sections of each experimental group were analyzed. Merged images of the DAPI and GFP signals are shown.

**Figure 5 animals-15-00128-f005:**
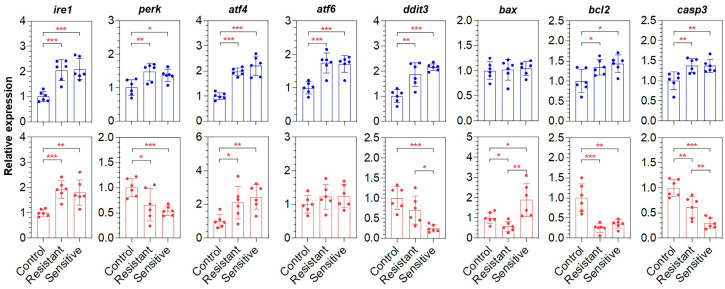
Expression of the endoplasmic reticulum (ER) stress response and apoptosis-associated genes determined by qPCR. The data for the brain and liver are shown in blue and red, respectively. The error bars indicate the standard deviation (SD), n = 6. *, *p* < 0.05; **, *p* < 0.01; ***, *p* < 0.001.

**Figure 6 animals-15-00128-f006:**
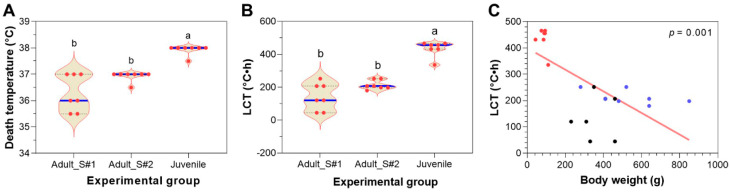
Comparison of the heat resistance of LMB adults and juveniles. (**A**) Death temperature of the fish (n = 7). (**B**) LCT of the fish (n = 7). The error bars indicate the standard deviation. Different letters above the error bars indicate a significant difference between the means (*p* < 0.05). (**C**) Correlation between the LCT and body weight of the fish. The black, blue and red dots indicate the individuals of adult_S#1, adult_S#2 and juvenile, respectively.

**Figure 7 animals-15-00128-f007:**
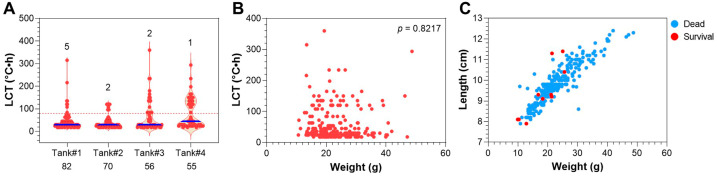
Selecting heat-resistant LMB juveniles with the tiered exposure method. (**A**) LCT of the experimental fish. The numbers of survivors are shown above the dots, the numbers of fish included in each replicate are shown below the tank number. The blue lines represent medians and the red dashed line separates heat-sensitive and resistant individuals. (**B**) Correlation between the LCT and body weight of the fish. (**C**) Correlation between the weight and length of the fish.

**Table 1 animals-15-00128-t001:** Source, development stage, body size and rearing temperature of the experimental fish.

Experiment Type	Experiment#	Experimental Group/Fish Source	Development Stage	Fish Number	Body Weight (g)	Standard Length (cm)	Rearing Temperature (°C)
CTmax	Experiment#1	NA	Juvenile	31	9.87 ± 3.14	7.78 ± 0.78	26–28
	Experiment#2	NA	Juvenile	19	15.65 ± 3.88	8.77 ± 0.74	26–28
Static exposure	Experiment#1	28, 30, 32, 34, 36 °C	Juvenile	10 fish per group	17.58 ± 5.00	8.38 ± 1.22	23–25
	Experiment#2	28, 30, 32, 34, 36 °C	Juvenile	20 fish per group	100.62 ± 23.31	15.89 ± 1.20	24–26
	Experiment#3	36.5, 37, 38 °C	Juvenile	10 fish per group	96.01 ± 17.53	15.54 ± 0.97	24–27
	Experiment#4	36, 36.5, 37 °C	Juvenile	20 fish per group	45.12 ± 10.47	12.32 ± 0.92	26–28
Tiered exposure	Experiment#1	Farm#1	Juvenile	23	59.89 ± 23.25	13.59 ± 1.51	26–28
		Farm#2	Juvenile	7	78.95 ± 25.55	14.83 ± 1.59	26–28
		Farm#3	Juvenile	20	87.83 ± 19.18	14.94 ± 1.16	26–28
	Experiment#2	Tank#1, Tank#2, Tank#3	Juvenile	33, 47, 48	23.66 ± 6.98	9.80 ± 0.81	25–28
	Experiment#3	Adult_S#1	Adult	7	364.29 ± 84.43	24.40 ± 1.91	26–28
		Adult_S#2	Adult	7	545.71 ± 184.56	27.19 ± 3.06	26–28
		Juvenile	Juvenile	7	84.06 ± 20.95	14.97 ± 1.42	26–28
	Experiment#4	Tank#1, Tank#2, Tank#3, Tank#4	Juvenile	82, 70, 56, 55	22.69 ± 6.89	9.96 ± 2.00	25–28

## Data Availability

The data presented in this study are available in this article.

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
