# Peer review of "Assessing Heat Resistance and Selecting Heat-Resistant Individuals of Largemouth Bass (Micropterus salmoides) with Tiered Thermal Exposure"

_animals, 2025, doi:10.3390/ani15020128_

Round 1

Reviewer 1 Report

Comments and Suggestions for Authors

The authors assessed the heat resistance of LMB, through two classical method and a new one. they also assessed transcriptomic responses of the fish to heat stress.

The study is interesting and has merits for publication. The following issues must be addressed:

- Abstract is good, but the authors should provide more details of the methods such as fish number, size, replication....

Introduction has focused on the global warming and the limitations of the classical methods of heat tolerance assessment. It should, however, be supplemented by information regarding the physiological and transcriptomic responses of the fish to heat stress to justify the study design.

In the methods, the authors should state the fish size and age.

The authors should provide citations of the classical methods of heat resistance assessment.

The authors should clearly indicate the number of fish sampled.

It is necessary to clarify how many sections were provided per sample for histological analysis. Also, the thickness of the sections should be stated.

statistical analysis is not clear. the authors should clearly indicate which test was used for each parameters.

The authors should state the name of statistical test and sample size for each figure/table in the results section.

Discussion was well-written.

Reviewer 2 Report

Comments and Suggestions for Authors

I had now time reading your manuscript entitled, " Assessing heat resistance and selecting heat-resistant individuals of largemouth bass (Micropterus salmoides) with tiered thermal exposure, which you submitted to the Animals.

I acknowledge that you have considerably revealed the interested data about the thermal biology of fish in the manuscript.

Abstract : The method of the tissue collection from this fish is still missing. Please add it. and Please specific tissue in Line 36.

Introduction: The introduction is still unclear about the thermal activity to induce the apoptotic cell in the animals and fishes. Why are the important of this knowledge? What is the benefits of the apoptotic cell assay? The review on these topics should be added and more information. It would be helpful to be more explicit about how this information could be used for this.

Why the brain and livers are important for this study ?
Although the author reported the sampled fish biology in Line 63-65, the importance of largemouth bass (Micropterus salmoides) is missing. More explanations about it !

Methods
Please the provide the analysis on the courting number of apoptotic cell in liver and brain. They are not clear state. Also, please show the previous references to support the method.

Results and discussion

Several errors and the academic presentations are still not detailed enough, in part incorrect, and unclear especially 3.4 Heat-sensitive fish demonstrated severe tissue damage and apoptosis. This is reflected to incorrect the results and discussion parts.

The authors should show the clearly detailed features between the content and figure 4. I saw Fig. 4A brain. What is the brain area that the author used? I think that this is the cerebellum, and it consists of an outermost molecular layer, a medial Purkinje cell layer, and an internal granule cell. However, this study is not revealed. Other brain regions should be also added.

Also, I saw the vacuolated structure of brain and liver in control? why? Is this the same position of brain between the experimental groups?

I sure that Fig. A: Liver – control is not the normal liver. The hepatic sinusoid and hepatocytes should be explained in Fig. 4.

I think that the localization of apoptotic cell should be the consecutive section from the Hand E staining method. Pls re-consideration! This is very good academic results between Hand E and TUNEL assay.

According to the presentation of the histopathological alterations, I suggested that the histological alteration index (HSI) is used. This method is a very effective tool for fish health situation. Please apply it to this study.

Comments on the Quality of English Language

The English could be improved to more clearly express the research.

Reviewer 3 Report

Comments and Suggestions for Authors

The manuscript studied thermal tolerance of largemouth bass using a variety of methods to determine heat tolerant and sensitive individuals. The authors used CTmax, static exposure and tiered thermal exposure method which is a novel approach to evaluating thermal performance. The authors also look at. This work contributed to exploitation of a new metric called Lethal cumulative temperature (LCT) in Aquaculture. On the whole, the manuscript was well organized and written, but requires addressing the following defects.

1. Why couldn't the CTMax and static exposure methods efficiently and accurately distinguish the heat resistance of LMB individuals?

2. What are the advantages of the tiered exposure method?

3. Why did you choose LCT as an indicator of heat resistance?

4. Why was the temperature increased at two different rates?

Reviewer 4 Report

Comments and Suggestions for Authors

The importance of environmental temperature in determining the physiology and growth of fish is well-known. Moreover, breeding heat-resistant fish, including largemouth bass, is of big significance in the face of global warming. The manuscript entitled «Assessing heat resistance and selecting heat-resistant individuals of largemouth bass (Micropterus salmoides) with tiered thermal exposure» by Haijie Chen et al. reports data on the classification of largemouth bass as sensitive or resistant to the heat stress based on the tiered exposure method, where temperature was increased in a stepwise manner and heat resistance of the fish was quantified as the cumulative lethal temperature. It is evident that the experiments reported are methodically designed, thus providing novel data. Subject to my evaluation, the sampling and experimental procedures appear to be adequate. The data are summarised in figures and tables; the quality of the presentation is satisfactory. The main finding is that largemouth bass juveniles are more resistant than the adults. Moreover, the livers of largemouth bass who were sensitive to heat showed more signs of tissue damage and cell death than the livers of the resistant ones. The text is clearly written. I would advise that the article be checked by a native English speaker. In general, the  MS is suitable for publication in ‘Animals’.

Round 2

Reviewer 2 Report

Comments and Suggestions for Authors

All contents in the manuscript are suitable for publications.